# Effect of (Pr+Ce) Additions on Microstructure and Mechanical Properties of AlSi5Cu1Mg Alloy

**Miao-Miao Fang [1,2], Hong Yan [1,2,]\*, Xian-Chen Song [1,2] and Yong-Hui Sun [1,2]**

1   School of Mechanical Electrical Engineering, Nanchang University, Nanchang 330031, China;
    15779575106@163.com (M.-M.F.); sxc.jl@163.com (X.-C.S.); 18720991493@163.com (Y.-H.S.)
2   Key Laboratory of Light Alloy Preparation & Processing, Nanchang University, Nanchang 330031, China
\*   Correspondence: hyan@ncu.edu.cn; Tel.: +86-791-8396-9633; Fax: +86-791-8396-9622

**Abstract:** The microstructure and mechanical properties of AlSi5Cu1Mg alloy with (Pr+Ce) addition were investigated by optical microscopy (OM), energy dispersive spectroscopy (EDS), and scanning electron microscopy (SEM). The results demonstrated that the rare earth (Pr+Ce) addition refined the grain. The long needle-like eutectic Si phases turned to granual. The secondary dendrite arm spacing (SADS) of the primary α-Al phase with the AlSi5Cu1Mg+0.6 wt.% (Pr+Ce) alloy reached the minimum value, which decreased by 50.2%. The mean length and the aspect ratio of the eutectic Si decreased by 78.8% and 67.4%. The ultimate tensile strength (UTS), the microhardness, and the breaking elongation of the AlSi5Cu1Mg+0.6 wt.% (Pr+Ce) alloy reached a maximum, and increased by 21.5%, 21.7%, and 8.0% compared to the AlSi5Cu1Mg alloy. The fracture examinations manifested in cleaved surfaces and brittle fracture areas, which were seen from the AlSi5Cu1Mg+0.6 wt.% (Pr+Ce) alloy. The number of dimples slightly increased.

**Keywords:** (Pr+Ce) modification; AlSi5Cu1Mg alloy; microstructure and properties; fracture

## 1. Introduction

The Al–Si alloy, which accounts for 80% of the aluminum alloy market, has many advantages including excellent castability ratings, high fluidity, and good corrosion resistance, etc. However, it is difficult to branch into additional application fields, due to the coarse microstructures. The Al–Si alloys were strengthened by the additional metal elements, such as Tungsten [1]. Wu et al. [2] considered the effects that Sr additions had on the microstructures and the mechanical properties of Al3Ti/ ADC12 (Aluminum-Alloy Die Castings) composites. They found that when the Sr content increase to 0.25 wt.%, the coarse dendritic of the primary α-Al phases completely refines, and the acicular or short rod shape of the eutectic silicon changes into a granular shape. The mechanical properties are improved by the addition of Sr, where good UTS and breaking elongation were obtained. Studies [3–7] show that some rare earth (RE) elements play a role in modifying the Al–Si cast aluminum alloys. Song et al. [8] investigated the impact that La had on the microstructure and hot crack resistance of an ADC12 alloy. The morphology of the α-Al grain changes from a dendritic crystal into fine equiaxed or spheroidal crystals when the quantity of La reaches 0.6 wt.%, and the size of the eutectic silicon varies from the needle-like to a tabular shape into fine rod-like shape without edges and corners. During the modification process of rare earths, the performance of aluminum alloys change by the different rare earth elements or processing techniques. Studies show that the addition of rare earth minerals modify the Fe phase to less harmful forms for Al–Si alloys [9]. Rao et al. [10] investigated the effects that Sm additions had on the eutectic Si, β-Al5FeSi phases of the ADC12 as-cast alloy. When the Sm addition increased to 1.0 wt.%, the needle-like β-Fe modifies into the Chinese script or the spherical α-Fe phase.

Previous studies [2–10] regarding the modification of the Al–Si alloy by mono-rare earth elements are informative. The similarity effect that the mixed rare earths had on the Al alloys is known, which alleviates the restrictions with solubility for the mono-rare earth elements, and the formation of the coarser intermetallic compounds is avoided, which improves heterogeneous nucleation. Zhu et al. [11] found that the sizes of the primary $\alpha$-Al phase and the eutectic Si phase were reduced as the (La+Ce) mass fraction increased in the range between 0.1 wt.% ~ 0.3 wt.%. Li et al. [12] investigated the effects that the (La+Yb) addition had on the microstructure and the tensile properties of the AlSi10Cu3 alloy. They detected that, the primary $\alpha$-Al phase and the acicular $\beta$-Al5FeSi phases were obviously refined because of the (La+Yb) addition, where the coarse acicular eutectic Si phases were modified into fine particles or a short rod morphology.

There are two hypotheses regarding the modification mechanism of the eutectic Si that rare commonly recognized. One view is that modifiers are forced to 'poison' Si and trigger the multiply twinning (IIT theory) [13]. There is also a TPRE (Twin Plane Reentrant Edge) mechanism [14] view that states the modifier can poison the re-entrant of the twin grooves and changes the morphology of the eutectic Si. The optimal ratio of the atomic radius of the modifier with regard to that of Si (Rmodifier/Rsi) to induce IIT is about 1.646 [13]. Pr and Ce have, atomic radius ratios of Pr/Si and Ce/Si that were 1.573 and 1.556. These values are close to the optimal atomic radius ratio. The purpose of this study is to investigate the (Pr+Ce) addition effect on the AlSi5Cu1Mg alloy's microstructures and the mechanical properties. The mechanism of the (Pr+Ce) modification is discussed.

## 2. Experimental

The AlSi5Cu1Mg-x (Pr+Ce) (x = 0, 0.3 wt.%, 0.6 wt.% and 0.9 wt.%) alloys were prepared from AlSi5Cu1Mg alloy and Al-5Pr-5Ce ternary master alloy. Table 1 shows the chemical compositions of the tested alloys were analyzed by an inductively coupled plasma-atomic emission spectroscopy. The prepared alloy was placed into preheated graphite crucible and melted completely in an electric resistance furnace at 720 °C. The Al-5Pr-5Ce master alloy was then added to the liquid to fabricate the AlSi5Cu1Mg-x (Pr+Ce) (x = 0, 0.3 wt.%, 0.6 wt.%, and 0.9 wt.%) alloy. The liquid was isothermally maintained at 720 °C for about 60 minutes to ensure that the Al-5Pr-5Ce master alloy was completely melted. The liquid was poured into a steel mold, which was preheated to about 180 °C.

**Table 1.** Chemical composition of AlSi5Cu1Mg-x(Pr+Ce) composites (mass fraction %).

| Number | Materials | Si | Cu | Mg | Fe | Pr | Ce | Al |
|--------|-----------|------|------|------|------|------|------|-------|
| 1 | AlSi5Cu1Mg | 5.41 | 1.05 | 0.50 | 0.63 | 0 | 0 | 92.41 |
| 2 | 0.3wt.% (Pr+Ce)/AlSi5Cu1Mg | 5.23 | 1.03 | 0.50 | 0.61 | 0.14 | 0.16 | 92.33 |
| 3 | 0.6wt.% (Pr+Ce)/AlSi5Cu1Mg | 5.21 | 1.01 | 0.50 | 0.60 | 0.32 | 0.28 | 92.08 |
| 4 | 0.9wt.% (Pr+Ce)/AlSi5Cu1Mg | 5.26 | 0.99 | 0.50 | 0.62 | 0.44 | 0.46 | 91.73 |

The metallographic specimens were etched with a 0.5 vol.% Hydrofluoric acid water solution. An optical microscope (OM, Nican M200 microscope) and a scanning electron microscope (SEM, FEI Quanta 200F) equipped with energy disperse spectroscopy (EDS, JSM-6701F) were used to characterize the microstructural evolution of the alloys. The grain size of Si was measured on Image-Pro Plus (IPP) software, the average length and aspect ratio according to the following Equations (1) and (2) [15]:

$$ML = \frac{1}{m} \sum_{j=1}^{m} \left( \frac{1}{n} \sum_{i=1}^{n} L_i \right)_j \tag{1}$$

$$AR = \frac{1}{m} \sum_{j=1}^{m} \left[ \frac{1}{n} \sum_{i=1}^{n} \left( \frac{l_l}{l_s} \right) \right] \tag{2}$$

*ML*: average length; *AR*: aspect ratio; $l_l/l_s$: ratio of length and width of the particle; *Li*: the length of a particle; *n*: the number of particles of a single field; and *m*: the number of the fields. Three test bars were fabricated for each alloy composition. The casting test bars were processed according to standard of GB/T 6397-86. The tensile test was implemented at 0.02 mm/min on an UTM5150 machine at the room temperature to measure the UTS and breaking elongation values. The fractured surfaces of the tensile test samples were examined via SEM. Vickers hardness of the casting samples were obtained by using a Vickers-hardness 1000 A hardness tester device (Laizhou Huayin Testing Instrument Co., Ltd., Laizhou, China) and at least six independent measurements were obtained.

## 3. Results and Discussion

### 3.1. Effects of (Pr+Ce) Addition on Microstructure and Morphology

The microstructures of the AlSi5Cu1Mg alloys with different contents of the (Pr+Ce) are shown in Figure 1. The Fe-containing phases (in Figures 1 and 2) of the AlSi5Cu1Mg alloy consisted of the Y-Al3FeSi, Al4Fe, β-Al5FeSi, α-Al8Fe2Si, and Al12FeMnSi phases [16]. As shown in Figure 3a, the eutectic Si phase in the matrix was a long-needle-like morphology. When 0.3 wt.% (Pr+Ce) was added, the eutectic Si phase was refined to a certain extent, the dendrites slowly disappeared, as shown in Figure 3b. The content of the mixed rare earth elements was insufficient; thus, the modification was incomplete and the eutectic Si phase required further refinement. When the addition of (Pr+Ce) increased to 0.6 wt.% (as shown in Figure 3c), The short rod-like of the eutectic Si phase was replaced by the fine granule of the eutectic Si phase structure. When the addition of (Pr+Ce) increased to 0.9 wt.%, the eutectic Si phase became coarser on some level (in Figure 3d). The size of Fe-containing phases were slightly larger than the phases in Figure 3b. The additional (Pr+Ce) had a significant modification effect on the AlSi5Cu1Mg alloy.

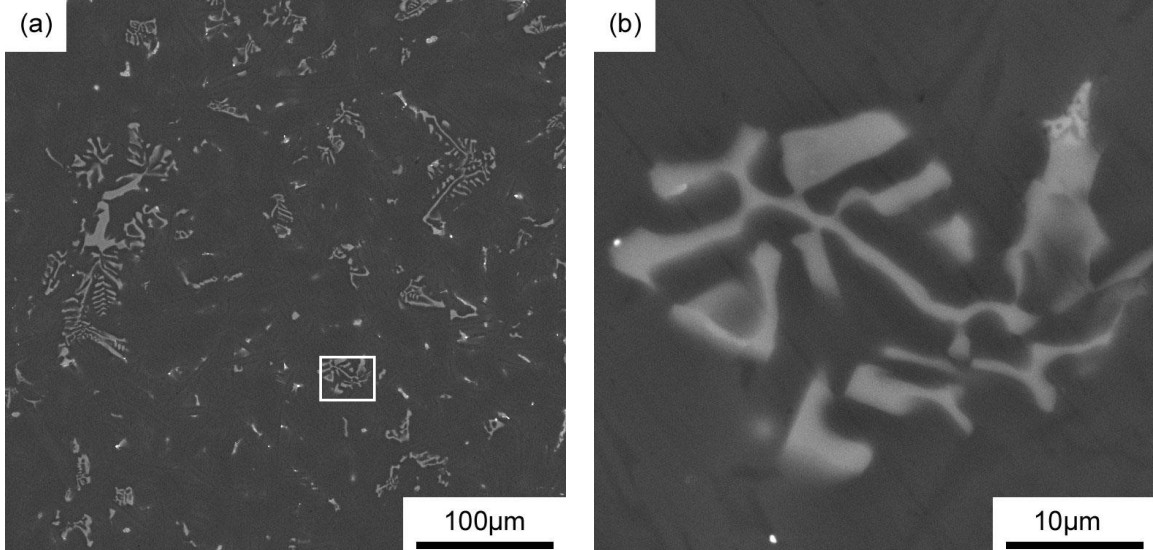

**Figure 1.** *Cont.*

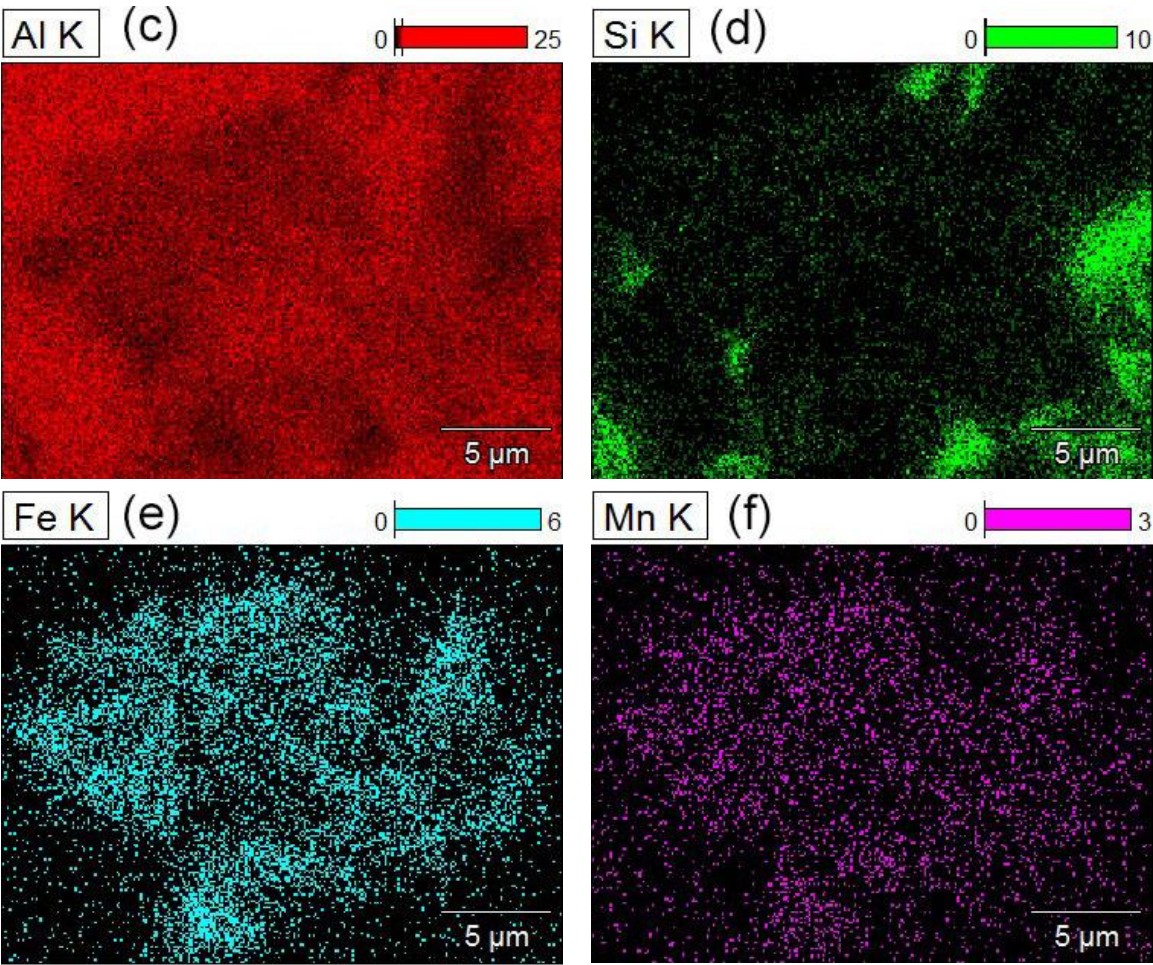

**Figure 1.** Backscattered SEM images of AlSi5Cu1Mg (**a**) at low; (**b**) high magnification and area scanning of AlFeMnSi phases of elements; (**c**) Al; (**d**) Si; (**e**) Fe; (**f**) Mn.

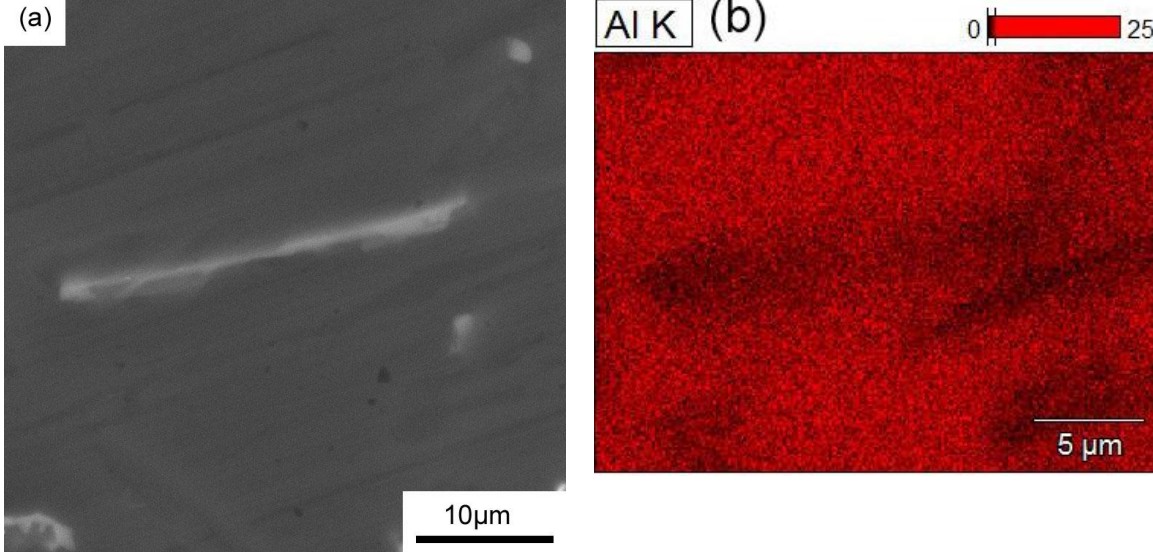

**Figure 2.** *Cont.*

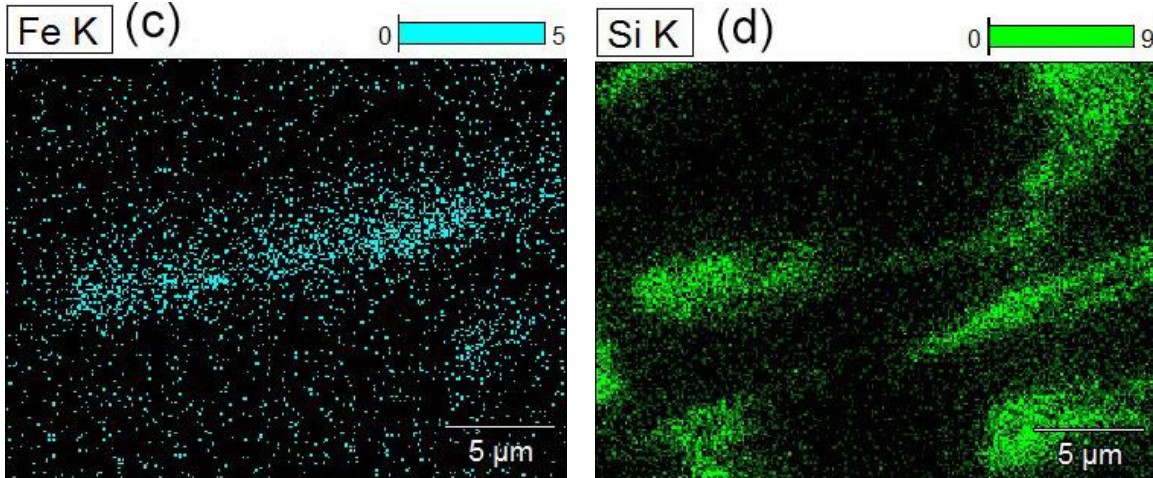

**Figure 2.** Backscattered SEM images of AlSi5Cu1Mg (**a**) high magnification and area scanning of AlFeSi phases of elements; (**b**) Al; (**c**) Fe; (**d**) Si.

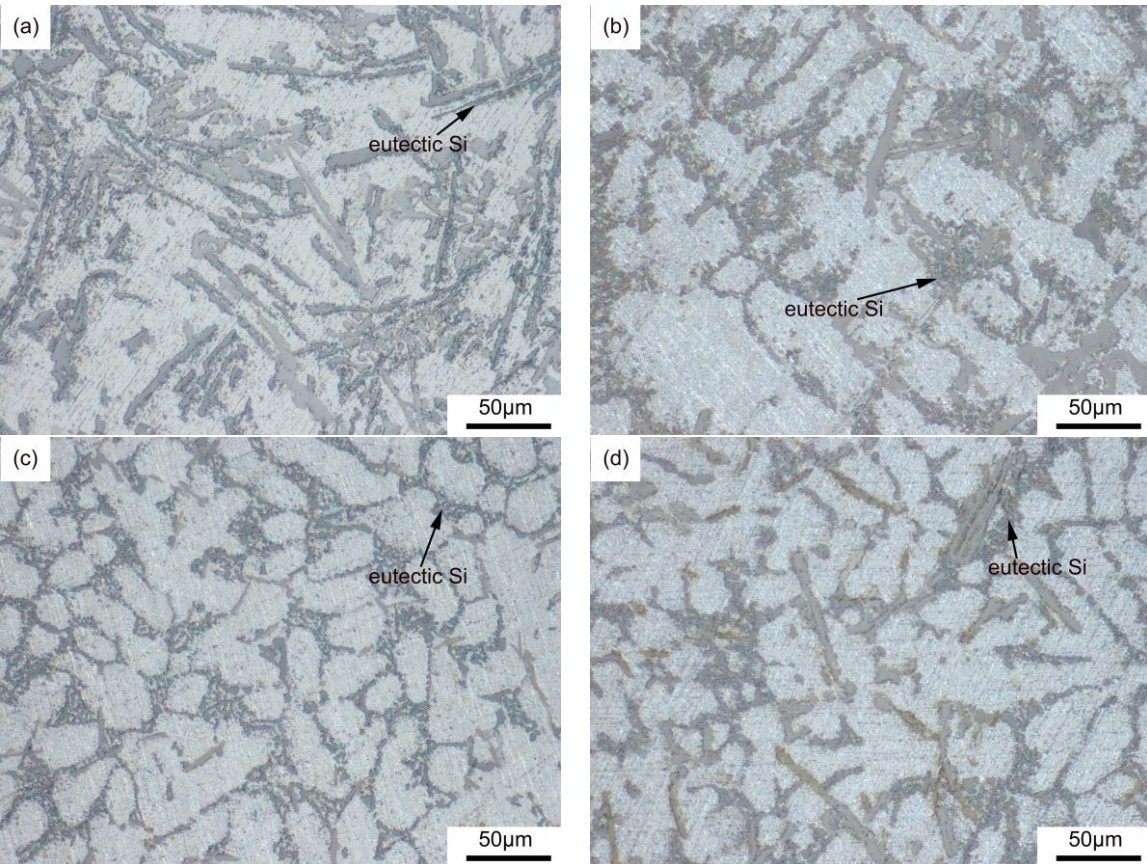

**Figure 3.** Microstructures of AlSi5Cu1Mg alloy with different (Pr+Ce) content: (**a**) 0; (**b**) 0.3 wt.%; (**c**) 0.6 wt.%; (**d**) 0.9 wt.%.

Figure 4 shows the change that various additions of the (Pr+Ce) caused on the dimension of eutectic Si. The mean length and aspect ratio were 15.1 μm and 10.6 for the matrix. When the additional (Pr+Ce) increased to 0.6 wt.%, the mean length and the aspect ratio were 3.2 μm and 3.5, which decreased by 78.8% and 67.4%. When the additional (Pr+Ce) increased 0.9 wt.%, the mean length and aspect ratio increased to 8.2 μm and 6.4.

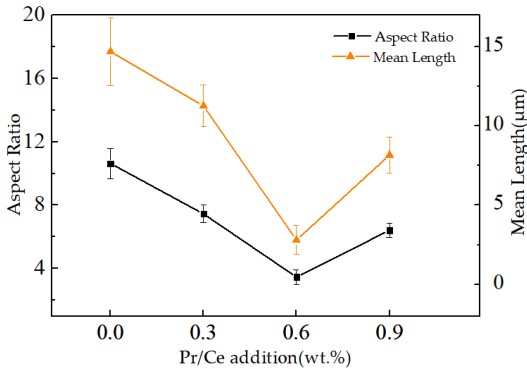

**Figure 4.** Dimensional of Si phases with different (Pr+Ce) additions.

The SDAS (Secondary dendrite arm spacing) of the AlSi5Cu1Mg-x(Pr+Ce) (x = 0, 0.3 wt.%, 0.6 wt.% and 0.9 wt.% increasing amount) alloys are shown in Figure 5. The SDAS decreased initially and then increased with additional of (Pr+Ce). When the additional (Pr+Ce) increased to 0.6 wt.%, the SDAS decreased by 50.2% compared to the value of the matrix alloy. For the Al–Si alloys, a smaller SDAS itself reduced the elemental segregation and improved the microporosity [17]. A density between the surfactant film [8] between the particles and the solution formed, due to the addition of the rare earth elements. The grain growth was largely inhibited and the grain was refined.

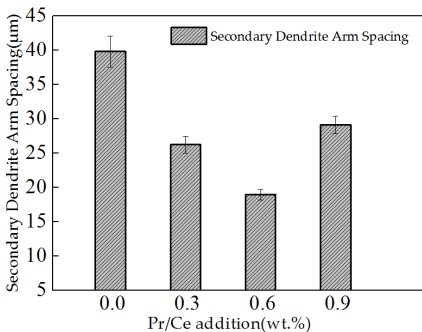

**Figure 5.** Value of SDAS of AlSi5Cu1Mg alloy with different (Pr+Ce) addition.

The eutectic reaction in the alloy occurred with the additional (Pr+Ce) [18]: $L \rightarrow \alpha$-Al + $Al_{11}RE_3$. A eutectic product ($Al_{11}RE_3$) was used as a nucleation core for $\alpha$-Al, which allowed grain refinement. Some studies [19] showed that many Al-rich phases (mainly including $Al_{11}RE_3$, $Al_3RE$, and $AlRE_3$) existed at the grain boundaries, which played an important role in inhibiting the grain boundary sliding and the matrix deformation, further improving the mechanical properties [20]. The results of XRD (X-ray diffraction) phase analysis in Figure 6 show that the addition of (Pr+Ce) can react with Fe-rich intermetallics compounds to form ternary Fe-rich intermetallics compounds of $Al_5CeFe_4$, which can improve the mechanical properties of large-size Fe-rich intermetallics relative alloys.

Based on scaling law $\lambda_2 = R/2$, the SDAS of value $\lambda_2$ for a small Peclet number factor was dependent on the following equation [21] (3):

$$\lambda_2 = \sqrt{\left(\frac{(8\Gamma DL)}{kv\Delta T_0}\right)} \tag{3}$$

$\lambda_2$: SDAS value; $\Gamma$: Gibbs Thomson coefficient; $D$: Diffusion coefficient in liquid; $L$: Constant depending on harmonic perturbations ranging from 10 to 28; $k$: Distribution coefficient; $v$: Dendritic front growth rate; $\triangle T_0$: Difference between the liquid and the solid equilibrium temperatures.

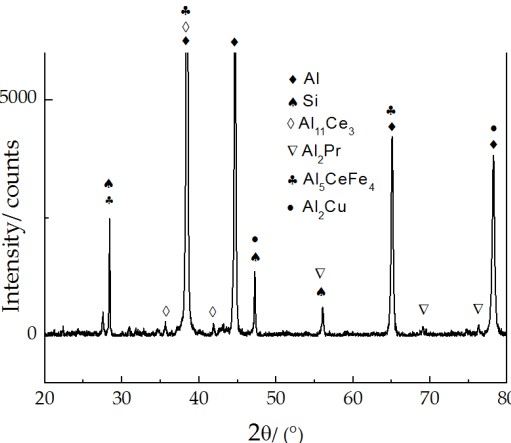

**Figure 6.** Result of the XRD analysis of AlSi5Cu1Mg-0.6 wt.% (Pr+Ce).

The Hume–Rothery principle [22] was added so that when the atomic radius difference between the solute and the solvent was more than 14%–15%, only low solid solubility formed in the alloy. Solid solubility of the rare earth element (Pr-0.182 um, Ce-0.184 um, Al-0.143 um) for Pr or Ce in Al matrix had a low possibility to enter the crystal lattice of the primary α-Al phase, instead gathering at the grain boundary [23]. During the solidification process, Pr and Ce aggregated at the solid–liquid interface and promoted the diffusion of solute atoms, further resulting in the composition undercooling, increasing the value of $\triangle T_0$, and decreasing the value of $\lambda_2$.

### 3.2. Mechanism of (Pr+Ce) Modification on Eutectic Si and Fe Phase

According to the TPRE mechanisms of the modified eutectic Si, the re-entrant edge with external corners of 141° acted as preferred growth sites and prevented the subsequent attachment of Si atoms, potentially driving the eutectic Si to preferentially along the growth direction, i.e., {112}$_{Si}$ [24]. The Al–Si system had a rival relationship between the growth of the Al phase and the Si phase. Interference with the nucleation and the growth of Al or Si could change the morphology of the Si [25]. The restricted nucleation theory states that when the temperature exceeded the eutectic temperature in the matrix, the free state of Si atoms existed in the liquid phase and the eutectic Si grew and nucleated [26]. The poisoning effect that the (Pr+Ce) addition rendered the Si phase in the grow mode of the Twin Plane Reentrant Edge (TPRE) [20]. This suggested that the Al-rich phases were preferentially absorbed on the growth interface of the eutectic Si phase, blocked the inherent growth step of the Si atom, and led to the growth surface of Si activity dropping. The eutectic growth stage combined with the restricted growth theories, so that the (Pr+Ce) intervened the growth tendency of Si and resulted in the eutectic Al growing before at the eutectic Si. Under these circumstances, the growth of Si passed through the channels between the Al cells with the help of the twin and formed the granual eutectic Si [27]. The Pr/Ce could not dissolve into the Al matrix. When the (Pr+Ce) was added in the melt, the matrix alloy provided a large number of nucleation points, which accumulated around the solid-liquid interface.

According to Li et al. [12], when RE (Rare earth) is added to the melt, parts of RE atoms will be absorbed on the surface of Fe-rich intermetallics and form a stable RE atom-coated membrane, which restrains the diffusion and precipitation of Fe atoms. Eventually, it will prevent the Fe-rich intermetallics from growing up. Furthermore, the Pr and Ce are materials with high surface activity. They are adsorbed on each crystal surface, changing the relative growth rate of each crystal surface during the crystal growth, promoting the dendritic dissociation and proliferation, and then changing the size of the grains. Figure 7 shows are backscattered SEM images of the alloy with AlSi5Cu1Mg-0.9 wt.% (Pr+Ce). The Al-rich intermetallics were attached on the surface of Fe-rich intermetallics compound to a certain extent, and the growth rate was slowed by reducing the surface energy difference in each crystal face, which improved the size of the Fe-rich intermetallics. The addition of rare earth elements

could provide a nucleation core of the Fe-rich intermetallics, which effectively increased the nucleation rate of the Fe-rich intermetallics and achieved a grain refinement effect.

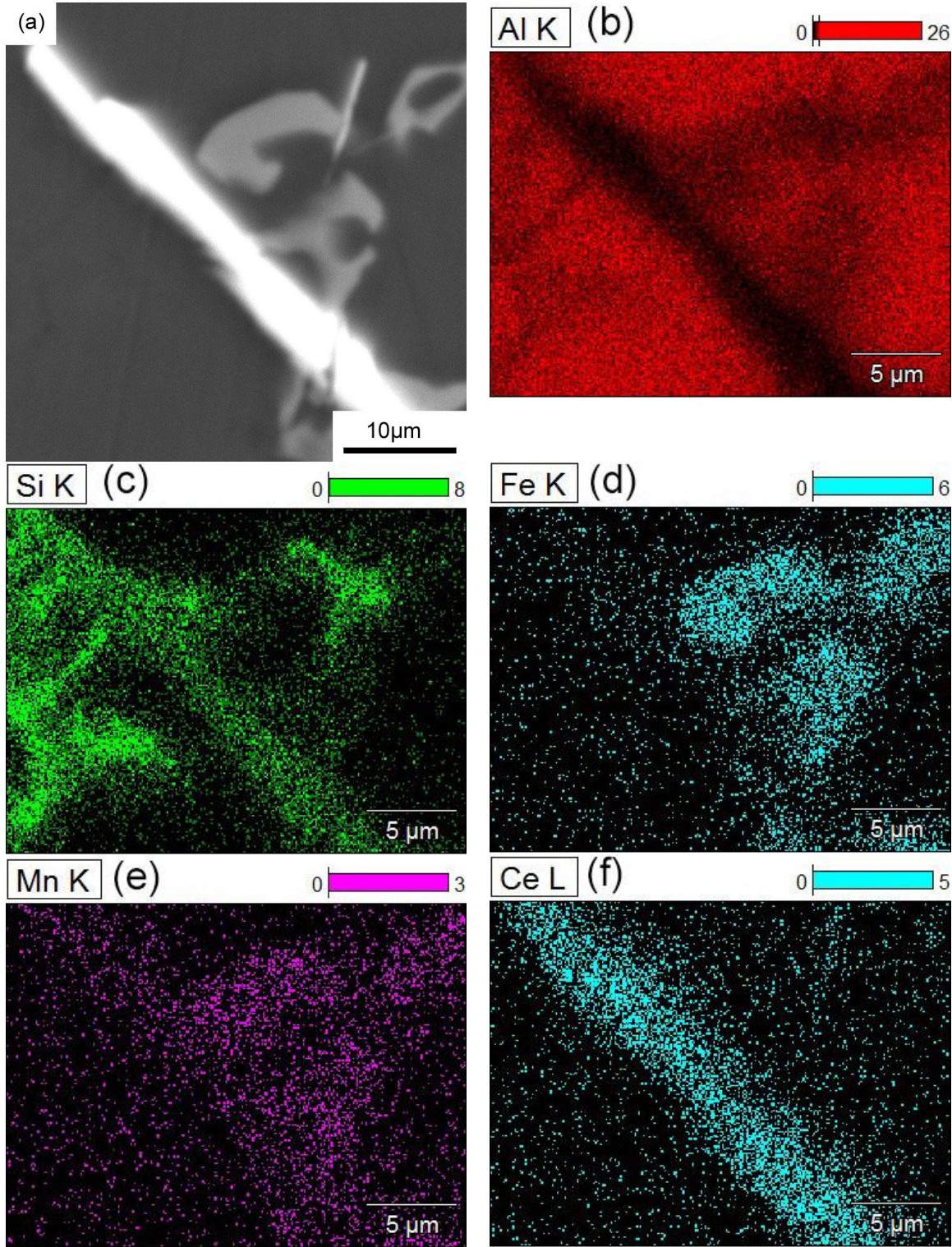

**Figure 7.** *Cont.*

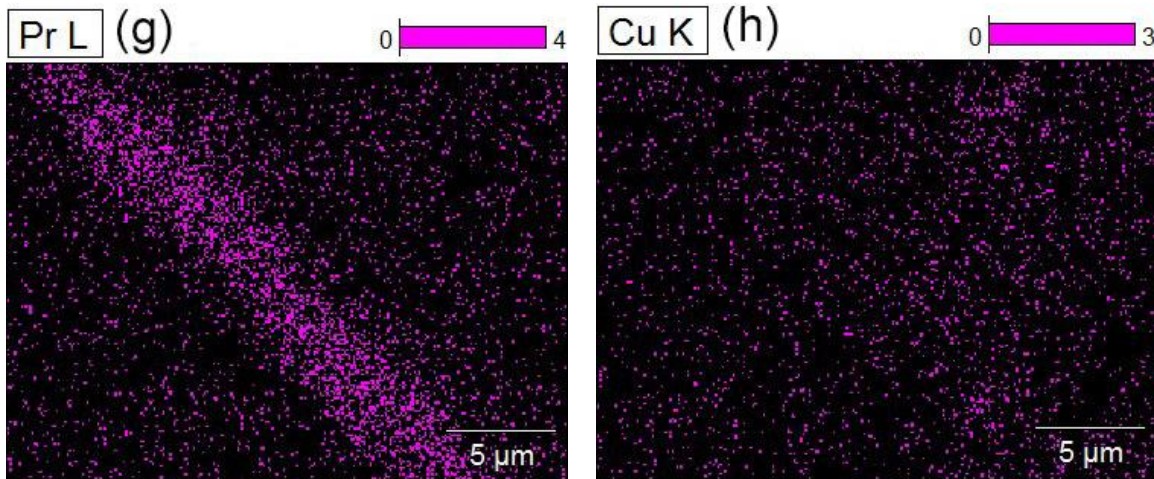

**Figure 7.** Backscattered SEM images of the alloy with AlSi5Cu1Mg-0.9 wt.% (Pr+Ce) (**a**) high magnification and area scanning of RE- rich (Rare earth-rich) intermetallic compounds of elements; (**b**) Al; (**c**) Si; (**d**) Fe; (**e**) Mn; f. Ce; (**g**) Pr; (**h**) Cu.

### 3.3. Effect of (Pr+Ce) Addition on Mechanical Properties

The UTS (ultimate tensile strength) and the breaking elongation of the AlSi5Cu1Mg with different (Pr+Ce) additions are shown in Figure 8. The UTS and the breaking elongation were remarkable elevated by the (Pr+Ce) addition, reaching 0-0.6 wt.%. The matrix alloy demonstrated the lowest tensile strength (230.5 MPa) and breaking elongation (5.4%). Among with the various additions of (Pr+Ce), AlSi5Cu1Mg-0.6 wt.% (Pr+Ce) exhibited a better UTS (280.4 MPa) and breaking elongation (5.8%). Compared to the matrix alloy, an increase of the UTS and the breaking elongation of 21.7% and 8.0% is achieved. When the addition of (Pr+Ce) increased to 0.9 wt.%, the mechanical properties of alloy declined gradually. Initially, (Pr+Ce) was separated at the grain boundary and strengthened the grain boundary. As additional (Pr+Ce) was added, the precipitated RE-rich phases acted as a dispersion strengthener. When the addition of (Pr+Ce) exceed 0.6 wt.%, the RE-enriched intermetallic compound accumulated in the front of the interface, and the grains became coarse by encroaching and overlapping, causing the mechanical degradation [18]. The mechanical properties of the cast Al–Si alloy were related to the morphology of the eutectic Si and Fe-containing phases [28]. The matrix alloy had many long-needle -like morphologies of the eutectic Si phases, bone-like or rod-like morphologies of the Fe-containing phases (in Figures 1 and 2), and the AlSi5Cu1Mg alloy exhibited lower mechanical properties.

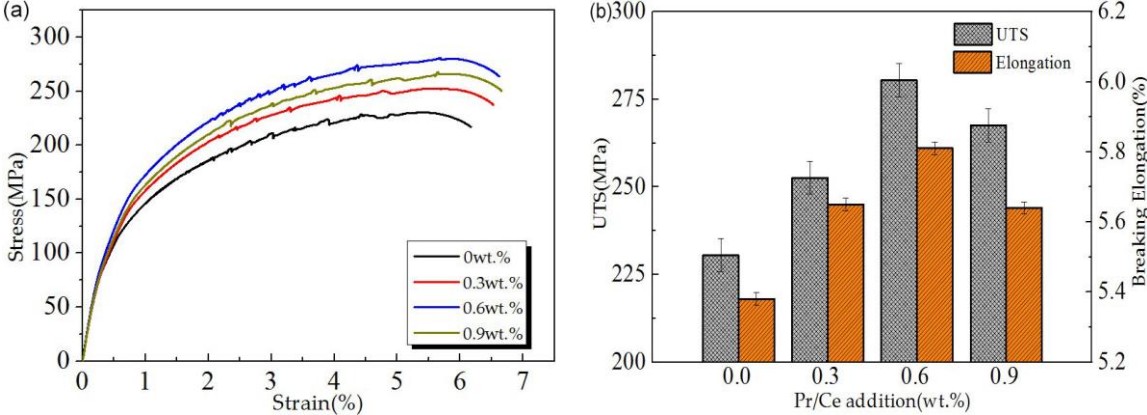

**Figure 8.** Tensile properties of AlSi5Cu1Mg alloy with different (Pr+Ce) additions (**a**) strain and stress; (**b**) breaking elongation and UTS.

Figure 9 shows a partial fracture surface of the AlSi5Cu1Mg alloy under various (Pr+Ce) additions. The fracture of the AlSi5Cu1Mg alloy was a typical cleavage fracture. Because the Si phases were long and needle-like, and it could easily generate internal stress concentration when it was subjected to an external force, causing cracking of the Si phase. The cracks at the adjacent Si phase were connected and gradually extended throughout the entire fracture surface, resulting in the alloy fracturing. Compared with the fractures of the matrix, the fracture of the 0.6 wt% (Pr+Ce) metamorphic alloy was mainly composed of dimples and a certain number of cleavage surfaces. The fracture mode was a quasi-cleavage fracture between the cleavage fracture and the dimple fracture. The internal grain boundary density of the cleavage fracture was relatively large, the intergranular interfacial bonding force was poor, and the grain boundary bonding force of the quasi-cleavage fracture was higher than the intergranular interfacial bonding force, resulting in the elongation of quasi-cleavage fracture being superior than the cleavage of the fracture. The grain size can have an important effect on the fracture mechanism of the alloy. As the addition of (Pr+Ce) increased further, most of the Si phase and Fe-containing phase become coarse, leading to the toughness and the breaking elongation of the alloy decreasing.

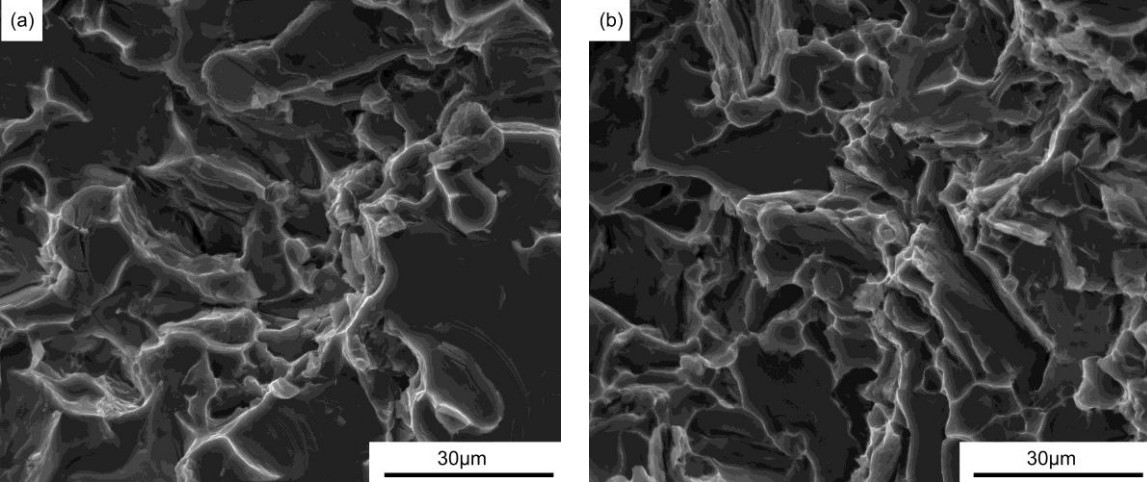

**Figure 9.** Fracture surfaces of the tensile samples (**a**): matrix; (**b**): 0.6 wt.% (Pr+Ce)

The grain size also has an important effect on the mechanical mechanism of the alloy, the smaller the size of the grain, the more grain boundaries. The dislocation motion between the grain boundaries was hindered. The alloy was strengthened and the tensile strength improved [29]. Besides, according to Hall–Petch Equation [30] (4):

$$\sigma_y = \sigma_0 + \frac{k_y}{\sqrt{d}} \tag{4}$$

$\sigma_y$: yield load (It can be expressed by microscopic Vickers hardness [31]); $\sigma_0$:materials constant for the starting load for dislocation movement (or the resistance of the lattice to dislocation motion); $k_y$: strengthening coefficient (a constant specific to each material); $d$: average grain diameter. The finer grain of the alloy and the higher strength resulted in the greater hardness of the alloy. The microhardness of the alloy is shown in Table 2. The microhardness of AlSi5Cu1Mg alloy was improved by adding (Pr+Ce). The alloy with 0.6 wt.% (Pr+Ce) had a highest microhardness of 87.9 HV, which increased 21.5% compared the matrix. In general, degassing and slag will reduce the porosity of castings. Around the pore formation of Al–Si–Cu alloy, many scholars have studied the porosity formation of Al–Si–Cu alloys. It is believed that the formation and distribution of porosity have a strong correlation with eutectic solidification mode [32–34]. However, there are many factors affecting eutectic solidification that need further study. Furthermore, some scholars believe that the addition of RE (≥3%) will increase the pore size [35], the increase of pore size will deteriorate the mechanical properties of

the alloy [36]. From Table 2 we can see that the appropriate amount of RE additions will not increase the porosity. This is because a small amount of rare earth is not enough to form coarse intermetallic compounds with alloy elements [37]. In addition, the addition of rare earth refines eutectic Si particles and Fe-containing intermetallic compounds, thus improving the mechanical properties of the alloy.

**Table 2.** Some properties of the samples used in the tests.

| Samples | Porosity/% | Microhardness/HV |
|---------|------------|------------------|
| AlSi5Cu1Mg | 0.55 | 72.39±3.15 |
| 0.3wt.% (Pr+Ce)/AlSi5Cu1Mg | 0.51 | 82.67±4.37 |
| 0.6wt.% (Pr+Ce)/AlSi5Cu1Mg | 0.49 | 87.96±3.48 |
| 0.9wt.% (Pr+Ce)/AlSi5Cu1Mg | 0.57 | 85.17±2.15 |

## 4. Conclusions

The (Pr+Ce) addition had a refinement effect on the eutectic Si. The morphology of the eutectic Si was modified from an acicular-like shape to a granular shape. When the (Pr+Ce) addition reached 0.6 wt.%, the mean length and the aspect ratio of the eutectic Si decreased by 78.8% and 67.4% compared to the matrix. The SDAS of the primary $\alpha$-Al phase diminished by 50.2%.

The (Pr+Ce) addition obtained a certain level of improvement in the mechanical properties of the AlSi5Cu1Mg alloy. The optimal addition of (Pr+Ce) was 0.6 wt.%, where the corresponding of microhardness, UTS, and breaking elongation increased by 21.5%, 21.7%, and 8.0%. The (Pr+Ce) addition could alter the fracture morphology of the alloy. The fracture mode of the 0.6 wt.% (Pr+Ce) metamorphic alloy was a quasi-cleavage fracture. The fracture appearances match the tendency of the tensile properties.

**Author Contributions:** M.-M.F., H.Y., X.-C.S. and Y.-H.S. conceived and designed the experiments; M.-M.F. and X.-C.S. performed the experiments; M.-M.F., H.Y., X.-C.S. and Y.-H.S. analyzed the data; M.-M.F., H.Y., X.-C.S. and Y.-H.S. contributed reagents/materials/analysis tools; M.-M.F., H.Y., X.-C.S. and Y.-H.S. wrote the paper.

**Funding:** This research was funded by the Natural Science Foundation of Jiangxi Province, grant number 20181BAB206026 and 20171BAB206034.

**Conflicts of Interest:** The author declares no conflict of interest.

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
