# Peer review of "Effect of (Pr+Ce) Additions on Microstructure and Mechanical Properties of AlSi5Cu1Mg Alloy"

_applsci, doi:10.3390/app9091856_

Round 1

Reviewer 1 Report

The article describes the influence of (Pr+Ce)-additions on the microstructure and mechanical properties of an AlSi5Cu1Mg Alloy. The Addition of (Pr+Ce) refines the grains and changes the form of the eutectic Si. The mechanical properties are enhanced by the (Pr+Ce) with an optimum at a content of 0.6%. The paper is well written except for a few exceptions. The suggestions for improvement are given in the added file.

Author Response

Thank you for the reviewers' comments concerning our manuscript.Those comments are all valuable and very helpful for revising and improving our paper. We have studied comments carefully and have made correction which we hope to meet with approval. Please find attached the main corrections in the paper and the responds to the reviewer’s comments.

Reviewer 2 Report

- The effect of (Pr+Ce) additions on microstructure and Mechanical properties of AlSi5Cu1Mg Alloy has been well studied.

-  The results obtained from the addition of (Pr+Ce) represents a certain level of improvement in the mechanical properties of the AlSi5Cu1Mg alloy.

-  SEM images in Fig (1) a,b,c and d were not captured in high enough magnification to justify the morphological changes in AlFeSi. A High magnification image in Inset to represent Plate like, Needle shaped, short rod-shaped and fine granule structure is suggested.

-  Pgae 3 line 95, correct the Fig. number.

-  Page 5 line 156, check the spelling for absorbed.

-  Higher magnification SEM image is suggested for Fig 4a to represent course needle like shape of eutectic Si phases and bone-like morphologies of the Fe containing phases.

-  In addition to spectrum of the elements, it is advisable to include the maps of the different elements to localize the presence of elemets.

Author Response

(The authors gave the same response as above.)

Reviewer 3 Report

Authors studied the effect of (Pr+Ce) on the microstructure and properties on Al-Si-Cu-Mg alloys. 

Though the study is interesting, but the results and presentation of figures, results and discussion is very weak to recommend this paper for publication. Some of the main criticisms are as given below:

Authors carried out particle analysis using image J and it is not clear how they were able to separate silicon particles from RE particles and Fe intermetallics. 

The quality of figures especially the microstructures need to be improved such as optical and SEM images and use proper micron markers for readers to read clearly. 

No XRD studies to determine the type of particles present in the alloy which would have increased the quality of the article. 

In the article, authors did not mention anything about the porosity formation and the effect of RE addition on the porosity formation and how the porosity can influence the mechanical properties. it is not clear whether they considered porosity in their study?

In the conclusions, it was mentioned that the size of AlFeMnSi type phase and AlFeSi phases were acceptable. It was not clear how this conclusion was made.

some typos: line #100; courser to coarser.

Line # 214. Fig 8 should be fig 7

Author Response

(The authors gave the same response as above.)

Round 2

Reviewer 3 Report

Authors made significant changes to the original article and now it may be accepted.